# Food Insecurity and Associated Factors Among Adolescents from Inland Northeast Brazil: A Cross-Sectional Study

**DOI:** 10.3390/ijerph22071087

**Published:** 2025-07-08

**Authors:** Maria Eliza Dantas Bezerra Romão, Maria Helena Rodrigues Galvão, Fábio Correia Sampaio, Jocianelle Maria Félix Fernandes Nunes, Franklin Delano Soares Forte

**Affiliations:** 1Program in Dentistry, Federal University of Paraíba, University City, João Pessoa 58051-900, PB, Brazil; elizamaria1-@hotmail.com (M.E.D.B.R.); mariahelena.galvao@ufpe.br (M.H.R.G.); fcsampa@gmail.com (F.C.S.); jocianelle@hotmail.com (J.M.F.F.N.); 2Public Health Center, Federal University of Pernambuco, University City, Vitória de Santo Antão 55608-680, PE, Brazil

**Keywords:** food insecurity, adolescent behavior, oral health

## Abstract

This study aimed to investigate food insecurity (FI) and its association with sociodemographic characteristics and behavioral and dental alterations in adolescents from a county in the inland of northeastern Brazil. Data on 192 adolescents aged 11–14 years were analyzed in the public school system in Juripiranga, Paraíba, Northeast Brazil. The adolescents and their guardians responded to the Brazilian Food Insecurity Scale, the Strength and Difficulties Questionnaire for the caregiver, the Strength and Difficulties Questionnaire for children and adolescents, and a sociodemographic questionnaire. Academic performance was observed by calculating the median of the final assessments of basic school subjects. The prevalence of FI was 69.19, and FI was associated with a family income of up to one minimum wage per month (prevalence ratio [PR]: 1.90; 95% confidence interval [95% CI]: 1.20–3.01), no practice of religion by the guardian (PR: 1.34; 95% CI: 1.04–1.73), behavior considered inappropriate by the guardian (PR: 1.33; 95% CI: 1.02–1.73), and academic performance (PR: 1.35; 95% CI: 1.05–1.72). FI is considered a complex and multifactorial problem that requires appropriate intervention to deal with multiple social determinants. The results point to the need for integrated public policies between the health, education, social assistance, and food security sectors.

## 1. Introduction

Access to safe and adequate food is essential to human life and is considered a determinant of health [1]. Multiple factors may change this condition and lead to food insecurity (FI) [2]. Food and nutritional security (FNS) is the exercise of the right of all citizens to regularly access quality food in an ideal quantity without hindering other essential needs [1,3]. The Brazilian National Household Sample Survey evidences an increased number of households in food and nutritional security [4]. This scenario was achieved by public policies aimed at guaranteeing FNS [3], such as the food acquisition program and the Bolsa Família Program [4].

FNS involves diverse and complex factors, since its behavior in a population depends on the availability, access, consumption, and use of food and nutrients [5]. Social groups with lower economic levels are exposed to instability because their income is more restricted [6]. Restricted budgets result in a reduced variety of foods and diets high in fermentable carbohydrates [7], which is concerning since the frequent consumption of high-fat and high-sugar foods causes systemic health issues and increases the risk of dental problems [8].

The expected nutritional development of children and adolescents is influenced by several factors, including gender, socio-economic status, physical activity, eating habits, religion, culture, and heredity [9]. In adolescence and childhood, food insecurity should be a public health concern and can have repercussions on various aspects of growth and development [10,11]. Nutritional disorders not addressed during formative years may persist into adulthood, which may lead to increased morbidities and decreased life expectancy [6]. The early identification of FI in adolescents at nutritional risk, combined with an understanding of family habits and behaviors, allows for the implementation of strategies during this crucial developmental stage and also facilitates the development of promotional actions that improve well-being [9].

Although the north and northeast regions of Brazil present more than half of all households having full and regular access to food, they report the lowest rates of food-secure private households (60.3% and 61.2%, respectively) [4]. In several households, these values correspond to 3.6 and 12.7 million in the north and northeast, respectively [4].

Studies conducted in large Brazilian urban centers have shown that food insecurity has a negative impact on various aspects of adolescents’ lives, including performance in daily and school activities, increased psychological stress, compromised nutritional status, and worsened quality of health [12,13,14]. A study carried out in the sertão region of Pernambuco found a high level of food and nutritional insecurity (moderate and severe forms were more frequent) associated with social conditions [15].

According to the Brazilian National Household Sample Survey of 2023, 35.9% of households in the state of Paraíba experience FI [4]. Understanding this reality is vital to support actions that favor the implementation of public policies related to food and nutrition, benefiting adolescents and their families [6]. The interaction between these factors is complex, indicating a close relationship between food insecurity, general health, sleep quality, eating patterns, and the physical and cognitive development of adolescents [11,15,16,17]. It is also considered that adolescents in a situation of food insecurity may show alterations in oral health parameters. In addition, the results available in studies carried out in large urban centers do not necessarily reflect the reality experienced by adolescents living in small municipalities, which have different socio-economic and structural contexts and often greater limitations in access to health services and public policies. This study aimed to investigate FI and its association with sociodemographic and behavioral variables, religious practices, lifestyle, dental caries, and fluorosis in adolescents from the inland of Paraíba.

## 2. Materials and Methods

### 2.1. Ethical Requirements

This study was approved by the research ethics committee of the Health Sciences Center of the Federal University of Paraíba (no. 53901321.0.0000.5188), following the Declaration of Helsinki and resolution 466/2012 of Brazil’s National Health Council. All legal guardians signed an informed consent form, and the participants signed an assent form.

### 2.2. Study Design and Sample Size

This cross-sectional study was conducted in Juripiranga, a county located in the northeast region of the state of Paraíba (in the Mata Paraibana mesoregion). The study comprised a larger project entitled “Oral health status of schoolchildren and associated factors in a small county in Paraíba”. According to the Brazilian Institute of Geography and Statistics, Juripiranga has a territorial extension of 78.706 km^2^, 10,012 inhabitants, a literacy rate of 72.9%, a Human Development Index of 0.548, and a Gini coefficient of 0.4210. This study focused on adolescents from Juripiranga since adolescence is a stage of social construction, behavior, and habits [18].

The adolescents were selected according to data from the municipal education department, which revealed a total of 500 adolescents between the ages of 11 and 14 enrolled. All adolescents enrolled in Juripiranga’s municipal education network were invited to take part in the study and were given all the terms and questionnaires that were part of this study.

G*Power software version 3.1.9.7 (Franz Faul, Universitat Kiel, Kiel, Germany) was used to determine the sample size, adopting a significance level of 95%. The study population comprised 500 adolescents aged between 11 and 14 years from public schools. With a finite population model of 500, 5% type I error rate, and 50% prevalence (due to ignorance of the phenomenon), the estimated sample size was 218 adolescents. Participants were selected according to the list of teenagers enrolled by the municipal education secretary [19].

### 2.3. Data Collection

Data were collected between February and November 2023 based on questionnaires applied to the adolescents and sent to guardians, along with a clinical examination of the oral cavity.

The questionnaire was organized into three sections. The first comprised sociodemographic and religious factors, such as income, family structure, education of the guardian, religious practice, use of dental services, tooth loss, frequency of tooth brushing, perception of the oral conditions of the adolescent, quality of sleep, and the consumption of fruit and sugary drinks.

The second section presented the Brazilian Food Insecurity Scale (EBIA), designed to assess perceptions of families regarding access to food, directly measuring the perception and experience of FI and hunger at the household level [20,21]. The EBIA was adapted and validated for the Brazilian population [20,21,22] and recommended for use in population studies in adolescents [21,22]. The EBIA consisted of 14 items; only positive answers were counted (1 point each), and the final score corresponded to their sum [20]. FI was classified as food security (0 points) and mild (1 to 5 points), moderate (6 to 9 points), or severe insecurity (10 to 14 points) [20].

The behavioral performance and mental health of the adolescents were assessed using two versions of the Strength and Difficulties Questionnaire (SDQ). The first was answered by the guardians (SDQ-R) and the second (SDQ-CA) by the adolescents; both versions were self-reported [23]. The SDQ consists of 25 items subdivided into five subscales. The first four provide the total of the difficulties (emotional symptoms, conduct problems, hyperactivity/attention conditions, problems with peers), and the fifth refers to positive behavior in the last six months, defined as pro-social behavior [23].

Each SDQ could be answered as false (score = 0), more or less true (score = 1), and true (score = 2). However, in five inverse items (7, 11, 14, 21, and 25), the score was false (score = 2), more or less true (score = 1), and true (score = 0). Since each of the five subscales includes five items or questions, the score for each subscale could vary from 0 to 10 [23].

The scores of the first four subscales generate a total of difficulties ranging from 0 to 40. The score for the prosocial behavior subscale was not included in the total score for difficulties, and the final score identified the cases as normal, borderline, or abnormal [23].

This study also examined the academic performance of adolescents by analyzing their final means in Portuguese, mathematics, history, geography, and science. The education secretary provided grades, and a median of the sample was calculated, which resulted in Md 8.00.

A single trained examiner performed the clinical examination of the oral cavity; adolescents underwent supervised brushing before this moment. The examination was conducted at school, with the participant seated, chin up, using artificial lighting, a clinical mirror, and a World Health Organization probe. The presence of dental cavities was observed using a jet of compressed air from a portable compressor.

The Thylstrup and Fejerskov index (1978) [24] was used to test dental fluorosis, which ranged from zero to three (no fluorosis) and four to nine (dental fluorosis).

The examiner was trained in two stages. The first was in lux, which involved reviewing images in a controlled setting. The second consisted of examining and re-examining 20 schoolchildren of the same age group who did not participate in the study, with a one-week interval to establish the Kappa (k) test. Inter-examiner agreement was k = 0.80 for dental cavities and k = 0.84 for dental fluorosis, and intra-examiner agreement was k = 0.80 for dental cavities and k = 0.88 for dental fluorosis.

### 2.4. Data Analysis

FI was the dependent variable categorized as with FI (score > 0) and without FI (score = 0), based on the EBIA scores. Poisson regression analysis with robust variance determined the prevalence ratios (PRs) and 95% confidence intervals (CIs) between FI outcome and variables, adjusting for potential confounding factors. Crude and adjusted analyses were performed using Poisson regression with a robust estimator to verify the estimated effect size and the precision of the association between the independent variables and food insecurity. Only variables with *p* < 0.20 (Wald test) in the crude evaluation were included in the multivariate model. Variables with *p* < 0.05 in the final model were considered significantly associated with the outcome. The results indicate an excellent fit: deviance goodness of fit = 59.00 (*p* = 1.000) and Pearson goodness of fit = 41.63 (*p* = 1.000).

## 3. Results

A cross-sectional survey was conducted among 192 adolescents aged 11–14 years, with a sample loss of 2.2% due to some participants dropping out. Food insecurity was present in 137 adolescents in this study, of whom 93 were mildly food insecure, 30 moderately food insecure, and 14 severely food insecure (Table 1).

With regard to sociodemographic characteristics, the majority of participants (79.2%) said that their family had a monthly income of up to one minimum wage, 93.6% of the adolescents said they had a father and/or mother as their guardian and 82.45% of the guardians had more than 8 years of schooling. The practice of religion was similar between the schoolchild (74.74%) and the carer (73.96%). Most adolescents (98.5%) and guardians (88.3%) considered their behavior to be appropriate. Clinical examination revealed a 42.9% prevalence of dental caries among the participants (Table 1).

Table 2 shows the results of the bivariate and multivariate tests of the logistic regression analyses regarding the relationship between the variables analyzed and food-insecure adolescents. In the final model, an income of up to one minimum wage (PR: 1.90; 95% CI: 1.20–3.01; *p* = 0.005), the non-practice of religion by the guardian (PR: 1.34; 95% CI: 1.04–1.73; *p* = 0.022), behavior considered inappropriate by the guardian (PR: 1.33; 95% CI: 1.02–1.73; *p* = 0.034), and grades below 8 (PR: 1.35; 95% CI: 1.05–1.72; *p* = 0.016) had a greater association with adolescents in situations with food-insecure adolescents (Table 2).

## 4. Discussion

### 4.1. Main Findings

This study was the first conducted in inland Paraíba state investigating FI among adolescents aged 11 to 14 years. National data showed that 21.4%, 7.4%, and 6.2% of households in Paraíba suffer from mild, moderate, and severe FI, respectively [4]. The prevalence of adolescents with FI was 69.19%, evidencing that most of the sample experienced hunger. This condition is considered a violation of the human right to adequate and necessary food [25]. FI was associated with an income of up to one minimum wage per month, corroborating previous studies reporting that low-income individuals have FI [6,25,26,27,28].

### 4.2. Interpretation of the Study Findings

Individuals with FI show reduced quality and variety in their diet [6,7,8,9,10,11,12,13,14,15,16,17,18,19,20,21,22,23,24,25]. They eat less good-quality foods (e.g., fruit, vegetables, proteins, and dairy products) and more foods of low nutritional quality (e.g., carbohydrates with large amounts of added sugars) [29].

A study conducted in Goiânia (Brazil) with adolescents aged from 12 to 18 found that 90% of the sample did not meet the intake of five portions/day of fruit, vegetables, and greens recommended by the World Health Organization. This finding is worrying since a low intake of these foods is one of the ten main risk factors for mortality worldwide [9]. Thus, the process that reduces the consumption of these foods must be further studied.

Studies have shown downward trends in food insecurity in some countries as a result of national initiatives and/or policies to combat food insecurity, with efforts around a Strategic Plan for the Development of Food and Nutrition, associated with an Education Sector Development Plan [11,30]. The same study showed an increase in food insecurity in other countries. Socio-economic difficulties and a lack of government support to deal with these issues may have contributed to this increase [11].

Low cost, greater accessibility in poorer places, and momentary satiety may justify the high consumption of foods of reduced nutritional quality by individuals with low incomes [6,9,31]. Families with higher incomes tend to consume high-quality foods, while the consumption of foods with low nutritional value is more common among individuals with low incomes [32]. Families facing financial difficulties are likely to reduce dietary costs, prioritizing the purchase of foods with high energy density and low nutrient content, which often leads to the poor quality of the diet [27,33]. Thus, individuals with income restriction, FI, and social vulnerability are exposed to instability in their quality of life [27,34].

The association between guardians of adolescents with FI and the absence of religious practice was an interesting result obtained in this study. The lack of prospects for people in social, economic, and food vulnerability may cause disbelief and detachment from collective practices that preach hope and a sense of well-being, such as religion [35]. Little or no religious practice may also be associated with a feeling of dissatisfaction with life, aggravated by low income and inadequate nutrition [36].

Understanding the negative impact of behavioral changes and impaired mental health on quality of life and family dynamics is essential [34]. Qualitative studies have suggested that FI affects many aspects of daily life, including situations apparently not related to food, such as social behavior [37,38], which justifies the association found in the present study between adolescents with FI and inadequate behavior from the perspective of guardians.

Moreover, individuals report difficulties in fulfilling social roles when concerned about what to eat and presenting behaviors considered adequate when socializing with family and friends [34].

Social relationships, behavior, and the development of activities in a collective environment are examples of issues that may impact the lives of adolescents with FI [38]. The prospect of improvement for families in vulnerable conditions is often placed in the academic development of children and adolescents [27,39].

A relationship was observed between FI and the academic performance of adolescents throughout 2023, whereby adolescents with FI presented lower means. The lack of adequate meals at home before going to school and the feeling of concern for the diet and future meals of the family are factors that may explain the lack of concentration in class and the poor performance of academic activities [11,17,27], school behavioral difficulties, and suspensions [15,40].

Changes in diet, such as the absence of nutrients for a prolonged period in childhood, may change the metabolism because the body adapts to contain energy expenditure, such as reduced body growth, neurological changes, and intellectual disorders [41]. Adolescents in this study may have experienced FI since childhood, which may have led to neurological and intellectual consequences, hindering learning in the traditional teaching context.

### 4.3. Policy Implications

Identifying adolescents with intellectual or neurological difficulties, promoting inclusion, developing ideal teaching methodologies, offering school meals of good nutritional quality, and seeking partnerships with the spheres of power to change the FI situation can minimize learning deficits and ensure food safety at school. Food insecurity in contexts of social vulnerability is a complex problem that requires integrated action from different sectors, such as education, health and social assistance, and employment and income generation, among others [15]. In order for intersectoral policies to be more effective in tackling this problem, some strategies can be adopted, such as expanding the National School Feeding Programme, effective monitoring by health professionals through the health at school program, and strengthening the single registry.

By integrating efforts between education, health, and social assistance, it is possible to build more effective and sustainable public policies to tackle food insecurity, ensuring that vulnerable populations have access to decent and nutritious food.

Most adolescents had previously visited a dentist. In Brazil, the Unified Health System (Sistema Único de Saúde—SUS) is founded on the principles of universality, equity, and comprehensiveness [42] and includes oral health as an integral component of care [43,44]. In the municipality of Juripiranga, primary health care is organized through the Family Health Strategy (FHS), which currently achieves 100% population coverage. FHS teams consist of physicians, nurses, nursing technicians, and community health workers. Complementing these are oral health teams—comprising dental surgeons and oral health assistants—who conduct ongoing health promotion, education, and disease prevention activities, in addition to delivering oral health care services aligned with the National Oral Health Policy (PNSB) [43,44]. The public water supply in the area contains naturally occurring levels of fluoride [45].

Moreover, the local population benefits from federal social protection programs, including the Bolsa Família Program (now incorporated into the Auxílio Brasil Program), as well as occasional municipal initiatives such as the distribution of basic food aimed at reducing food insecurity. These efforts align with and contribute to the achievement of the United Nations’ Sustainable Development Goal 2 (SDG 2), which aims to eradicate hunger and ensure access to safe, nutritious, and sufficient food for all by 2030 [46]. Above all, it is essential to strengthen strategies with the potential to mitigate the impacts of food insecurity on the well-being and holistic development of adolescents living in socially vulnerable contexts. Key actions include nutritional and growth monitoring in schools and health facilities, strengthening the National School Feeding Program (PNAE), individualized nutritional follow-up, and the training of health and education professionals to identify early signs of food insecurity and make appropriate referrals [47]. When implemented in a coordinated and territorially responsive manner, these interventions can significantly contribute to promoting equity and the holistic health of at-risk adolescents. Achieving this requires concerted efforts across the education, health, social assistance, and justice sectors [11,48].

### 4.4. Strengths and Limitations

A few limitations of this study must be addressed. Although the examiner was trained and the instruments were validated to ensure the reliability of the results, the cross-sectional design of this study hindered the determination of cause-and-effect relationships. These limitations also relate to the use of self-reported data, which may be influenced by recall bias and social desirability. Additionally, the findings are generalizable only to adolescents from small municipalities who attend public schools, as this was the specific population included in the study. Thus, the results of the present study may contribute to the reflection of food and nutrition actions and policies and their implementation for populations and communities in situations of social inequality.

## 5. Conclusions

A high prevalence of FI was observed in adolescents aged 11 to 14 years, evidencing a multifactorial problem that requires intervention in multiple social determinants. The present study also found an association between adolescents with FI and low income, inadequate behavior perceived by guardians, the absence of religious practice by guardians, and lower academic performance. There is an urgent need to implement integrated public policies that promote access to adequate and healthy food—such as the National School Feeding Program (PNAE), the Bolsa Família Program, and other social initiatives—while also reinforcing healthy eating practices in schools. These efforts should aim to enhance effective coordination among the health, education, social assistance, and food security sectors.

## Figures and Tables

**Table 1 ijerph-22-01087-t001:** Sociodemographic characteristics of the sample. Juripiranga, Paraíba, Brazil, 2023.

Variables	*n*	%
Sociodemographic		
Food insecurity		
*Yes*	137	69.19
*No*	61	30.81
Income		
*Up to 1 MW*	152	79.20
*More than 1 MW*	40	20.80
Guardian		
*Father or mother*	176	93.60
*Other*	12	6.40
Education of the guardian		
*No schooling*	26	13.83
*High school*	155	82.45
*Higher education*	7	3.72
Practice of religion (adolescent)		
*Yes*	145	74.74
*No*	49	25.26
Practice of religion (guardian)		
*Yes*	142	73.96
*No*	50	26.04
Perception of behavior (view of guardian)		
*Adequate*	113	88.30
*Inadequate*	15	11.70
Perception of behavior (view of adolescent)		
*Adequate*	135	98.50
*Inadequate*	2	1.50
Oral Health		
Visit a dentist		
*Yes*	185	93.40
*No*	13	6.60
Last appointment		
*Public*	149	80.5
*Private*	36	19.5
Tooth loss		
*Yes*	41	21.40
*No*	151	78.60
Dental cavities		
*Yes*	113	42.90
*No*	85	57.10
Dental fluorosis (more than TF 3)		
*Yes*	28	14.58
*No*	117	88.54
Behavioral variables		
Frequency of eating sugar		
*Always*	58	29.30
*Never*	140	70.70
Frequency of eating fruits		
*Always*	104	52.80
*Never*	93	47.20
Frequency of drinking sugary drinks		
*Always*	64	32.30
*Never*	134	67.70
Sleep quality		
*Good*	135	68.20
*Bad*	63	31.80
Access to piped water		
*Yes*	148	76.7
*No*	45	23.3
Academic performance		
Academic performance		
*Above 8.0*	118	61.8
*Below 8.0*	73	38.2

MW = minimum wage in reais for the year 2023 (BRL 1302).

**Table 2 ijerph-22-01087-t002:** Association between food insecurity and characteristics of adolescents using Poisson regression. Juripiranga, Paraíba, Brazil, 2023.

Variables	Food Insecurity				Crude PR (95% CI)	*p*-Value	Adjusted PR (95% CI)	*p*-Value
	Yes		No					
	*n*	%	*n*	%				
Income								
*Up to 1 MW*	35	23.0	117	76.0	1.81 (1.24–2.62)	0.002	1.90 (1.20–3.01)	0.005
*More than 1 MW*	23	57.5	17	42.5	1		1	-
Guardian								
*Father or mother*	51	28.9	125	71.0	1	-	-	-
*Other*	6	50.0	6	50.0	0.71 (0.64–0.78)	0.232	-	-
Education of the guardian								
*No schooling*	11	42.3	15	57.6	0.67 (0.42–1.05)	0.083	-	-
*High school*	45	29.0	110	70.9	0.82 (0.60–1.13)	0.247	-	-
*Higher education*	1	14.2	6	85.7	1	-	-	-
Practice of religion								
(adolescent)								
*Yes*	50	34.4	95	65.5	1	-	-	-
*No*	7	14.2	42	85.7	1.30 (1.10–1.54)	<0.001	-	-
Practice of religion								
(guardian)								
*Yes*	48	33.8	94	66.2	1	-	1	-
*No*	8	16.0	42	84.0	1.26 (1.07–1.50)	0.006	1.34 (1.04–1.73)	0.022
Perception of behavior (view of guardian)								
*Adequate*	40	35.4	73	64.6	1	-	1	-
*Inadequate*	3	20.0	12	80.0	1.23 (0.92–1.65)	0.147	1.33 (1.02–1.73)	0.034
Perception of behavior (view of adolescent)								
*Adequate*	47	34.8	88	65.1	1		-	-
*Inadequate*	0	0.0	2	100	1.53 (1.35–1.73)	<0.001	-	-
Visit a dentist								
*Yes*	59	31.8	126	68.1	1	-	-	-
*No*	2	15.8	11	84.6	1.24 (0.96–1.59)	0.092	-	-
Tooth loss								
*Yes*	42	27.8	109	72.1	0.74 (0.54–1.00)	0.05	-	-
*No*	19	46.3	22	53.6	1	-	-	-
Dental cavities								
Yes	26	39.5	59	69.4	0.99 (0.82–1.20)	0.954	-	-
No	35	30.9	78	69.0	1	-	-	-
Dental fluorosis (more than 3)								
*Yes*	20	14.1	8	4.0	1.13 (0.46–2.73)		-	-
*No*	117	59.1	53	26.8	1	0.486	-	-
Last appointment								
*Public*	39	26.1	110	73.8	1	-	-	-
*Other*	20	55.5	16	44.4	0.60 (0.41–0.87)	0.009	-	-
Frequency of eating sugar								
*Always*	20	34.4	38	65.5	0.92 (0.74–1.14)	0.488	-	-
*Never*	41	29.2	99	70.7	1	-	-	-
Frequency of eating fruits								
*Always*	33	31.7	71	68.2	1	-	-	-
*Never*	28	30.1	65	69.8	1.02 (0.84–1.23)	0.806	-	-
Frequency of drinking sugary drinks								
*Always*	22	34.2	42	65.6	0.92 (0.75–1.14)	0.468	-	-
*Never*	39	29.1	95	70.9	1	-	-	-
Sleep quality								
*Good*	43	31.8	92	68.1	1	-	-	-
*Bad*	18	28.5	45	71.4	1.04 (0.86–1.27)	0.636	-	-
Access to piped water								
*Yes*	50	33.7	98	66.2	1	-	-	-
*No*	8	17.7	37	82.2	1.24 (1.03–1.48)	0.017	-	-
Academic performance								
*Above 8*	44	37.2	24	62.7	1	-	1	-
*Below 8*	17	23.2	56	76.7	1.22 (1.01–1.47)	0.036	1.35 (1.05–1.72)	0.016

MW = minimum wage in reais for the year 2023 (BRL 1302).

## Data Availability

The datasets used and/or analyzed during the current study are available from the corresponding author upon reasonable request.

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
