# Peer review of "Food Insecurity and Associated Factors Among Adolescents from Inland Northeast Brazil: A Cross-Sectional Study"

_ijerph, 2025, doi:10.3390/ijerph22071087_

Round 1

Reviewer 1 Report

Comments and Suggestions for Authors

Review,

The authors present an extensive and complex study regarding Food insecurity and associated factors among adolescents from 2 the Inland Northeast Brazil

Although the study is very well structured and written after the evaluation, I can present the following aspects.

Comment 1

TITLE

I suggest that the title includes the type of study.

Comment 2

ABSTRACT

The abstract is well structured and contains important data but please add explanations to the line 27 for AI. Please differentiate each chapter in the abstract, namely, introduction, material and method, results and discussions, conclusions.

Comment 3

INTRODUCTION

The introduction provides the context of the study by providing data both nationally and internationally. Updating the information by using more recent bibliographic sources would considerably help the quality of the information presented. I suggest supplementing it with the results of more recent research. Please find a medical expression and eliminate dental alterations line 76.

Comment 4

MATERIAL AND METHOD

The study is described in detail, but the methodology is not very clear.  They are presented succinctly and completely with the materials and the methods of the study. I suggest a brief explanation in stages, first, the materials/tools used should be presented and then the method individualized for each stage.

Comment 5

Please pass the value of the Cronbach's Alpha index.

Comment 6

RESULT

They are presented succinctly and completely. Try not to start your sentence with a number, line 162.

Comment 7

DISCUSSIONS

In the discussion section, I suggest that they be individualized according to the results obtained and every aspect compared with the results of other studies.

The conclusions are clearly correlated with the data analyzed.

Comment 8

REFERENCES

The bibliography contains less than half of the sources from the last 5 years, respectively 28%. I suggest improving this aspect.

Author Response

We would like to thank the reviewers for their contributions and resend the article entitled ‘Food insecurity and associated factors among adolescents from the Inland Northeast Brazil’ with the suggested corrections. Firstly, express our gratitude to the reviewers for their valuable recommendations, questioning and reflections that contributed significantly to improving the content and writing of the manuscript.

In accordance with their recommendation to highlight the changes suggested by the reviewers in the by the reviewers in the manuscript using the track changes mode in MS Word or by using bold or colored text, in this revised version we have adopted colored text (in red) to indicate the changes made. red) to indicate the changes made.

The following justifications have been ordered following the comments of each reviewer:

Review 1,

The authors present an extensive and complex study regarding Food insecurity and associated factors among adolescents from 2 the Inland Northeast Brazil

Although the study is very well structured and written after the evaluation, I can present the following aspects.

Comment 1

TITLE

I suggest that the title includes the type of study.

Authors' response: We thank the reviewer for the suggestion. The change was made in the title

Comment 2

ABSTRACT

The abstract is well structured and contains important data but please add explanations to the line 27 for AI. Please differentiate each chapter in the abstract, namely, introduction, material and method, results and discussions, conclusions.

Authors' response: We thank the reviewer for the suggestion. The change was made sentence, change AI for FI. Observing the instructions to authors there is no obligation to namely each chapter: introduction, material and method, results and discussions, conclusions

Comment 3

INTRODUCTION

The introduction provides the context of the study by providing data both nationally and internationally. Updating the information by using more recent bibliographic sources would considerably help the quality of the information presented. I suggest supplementing it with the results of more recent research. Please find a medical expression and eliminate dental alterations line 76.

Authors' response: we thank the reviewer for the suggestion. We have added and verified that there are 10 references from the last 5 years. We have changed the expression dental alterations to caries and dental fluorosis.

Comment 4

MATERIAL AND METHOD

The study is described in detail, but the methodology is not very clear.  They are presented succinctly and completely with the materials and the methods of the study. I suggest a brief explanation in stages, first, the materials/tools used should be presented and then the method individualized for each stage.

Authors' response: important suggestion. The writing was guided by STROBE. The journal's guidelines do not require a separation between materials and methods. The method follows a sequence: ethical aspects, Study design and sample size, Data collection (describing where the data was collected, how it was collected and by whom and when it was collected), Data analysis (describing how the data was processed in order to achieve the proposed objectives). The writing also took into account the number of words according to the journal's guidelines.

Comment 5

Please pass the value of the Cronbach's Alpha index.

Authors' response: We appreciate the reviewer’s suggestion regarding the inclusion of Cronbach’s alpha to assess the internal consistency of our instrument. However, we would like to clarify that our analysis was not intended to validate a psychometric scale or a set of items measuring a unidimensional latent construct. Therefore, the assumptions underlying the use of Cronbach’s alpha—such as tau-equivalence and unidimensionality—are not met in this context. Nonetheless, we performed the calculation as requested, and the result was a negative Cronbach’s alpha (α = -0.077; standardized α = 0.016) across the 17 items. This negative value reflects a violation of the basic assumptions of internal consistency analysis, likely due to negative average inter-item covariances. Such a result further supports our position that Cronbach’s alpha is not appropriate for the current data set or research design. We hope this clarification addresses the reviewer’s concern and are happy to provide any additional information if necessary.

Comment 6

RESULT

They are presented succinctly and completely. Try not to start your sentence with a number, line 162.

Authors' response: We thank the reviewer for the suggestion. The change was made

Comment 7

DISCUSSIONS

In the discussion section, I suggest that they be individualized according to the results obtained and every aspect compared with the results of other studies.

Authors' response: We thank the reviewer for the suggestion. The change was made

Comment 8

The conclusions are clearly correlated with the data analyzed.

Authors' response: We thank the reviewer for the suggestion.

Comment 9

REFERENCES

The bibliography contains less than half of the sources from the last 5 years, respectively 28%. I suggest improving this aspect.

Authors' response: We thank the reviewer for the suggestion. We tried to improve by adding more references from the last five years.

Reviewer 2 Report

Comments and Suggestions for Authors

Thank you for submitting this well-structured and relevant manuscript. The study addresses an important public health issue by exploring the prevalence and associated factors of food insecurity among adolescents in a socially vulnerable region of Brazil. The use of validated tools and appropriate statistical methods strengthens the findings.

Minor revisions are recommended to improve clarity in the methods section, better emphasize the study’s novelty in the introduction, and refine the language throughout the manuscript.

Please refer to the detailed comments in the attached PDF for section-by-section suggestions.

Comments on the Quality of English Language

The English is generally clear and the manuscript is understandable. However, several sentences—particularly in the Introduction and Discussion—would benefit from improved grammar, more concise phrasing, and better flow. A light revision by a native English speaker or professional editor is recommended to enhance readability and precision.

Author Response

Review 2

Authors' response: We thank the reviewer for the suggestion

Comment Abstract

The abstract presents key findings but the conclusion is generic. Consider specifying witch social determinants require intervention

Avoid repetition in reporting prevalence and associated factors, condense for clarity

Authors' response: We thank the reviewer for the suggestion. we rewrite the conclusions

Comment Introduction

The rationale could be strengthened by briefly citing earlier studies on adolescent FI in Brazil for comparation

The final paragraph should better highlight the novelty of this study (e.g. focus on a small inland municipality, link with dental/ oral outcomes)

Authors' response: We thank the reviewer for the suggestion. We have added other studies as suggested and restructured the justification. Thank you very much.

Fares, K.; Barada, D.; Hoteit, M.; Abou Haidar, M. Prevalence and Correlates of Food Insecurity among Lebanese University Students of Hadath Campus. Atena J. Public Health 2020, 2, 5.

Smith L, López Sánchez GF, Tully MA, Jacob L, Kostev K, Oh H, Butler L, Barnett Y, Shin JI, Koyanagi A. Temporal Trends in Food Insecurity (Hunger) among School-Going Adolescents from 31 Countries from Africa, Asia, and the Americas. Nutrients. 2023 Jul 20;15(14):3226.

Valente CRM, Marques CG, Nakamoto FP, Salvalágio BR, Lucin GA, Velido LCSB, Dos Reis AS, Mendes GL, Bergamo ME, Okada DN, D Angelo RA, de Lázari EC, Dos Santos Quaresma MVL. Household food insecurity among child and adolescent athletics practitioners: A cross-sectional, descriptive, and exploratory study. Nutrition. 2024;123:112407. doi: 10.1016/j.nut.2024.112407.

de Amorim ALB, Dalio Dos Santos R, Ribeiro Junior JRS, Canella DS, Bandoni DH. The contribution of school meals to food security among households with children and adolescents in Brazil. Nutrition. 2022;93:111502. doi: 10.1016/j.nut.2021.111502.

Facina VB, Fonseca RDR, da Conceição-Machado MEP, Ribeiro-Silva RC, Dos Santos SMC, de Santana MLP. Association between Socioeconomic Factors, Food Insecurity, and Dietary Patterns of Adolescents: A Latent Class Analysis. Nutrients. 2023;15(20):4344. doi: 10.3390/nu15204344.

Santos NF et al. Excesso de peso em adolescentes: insegurança alimentar e multifatorialidade no cenário do semiárido de Pernambuco. Rev Paul Pediatr. 2020;38:e2018177

Kotchick, B.A.; Whitsett, D.; Sherman, M.F. Food Insecurity and Adolescent Psychosocial Adjustment: Indirect Pathways through Caregiver Adjustment and Caregiver–Adolescent Relationship Quality. J. Youth Adolesc. 2021; 50:89–102.

Baer, T.E.; Scherer, E.A.; Fleegler, E.W.; Hassan, A. Food Insecurity and the Burden of Health-Related Social Problems in an Urban Youth Population. J. Adolesc. Health. 2015; 57:601–607

Comment Methods

Study design: Clearly described. However, a clearer distinction between the larger project and the current sub-study would help.

Sample selection: the recruitment strategy is sound, but it´s unclear whether any stratification was applied. Consider clarifying.

Instruments: EBIA and SQD are properly explained. Still, consider citing validation studies in similar adolescent populations

Fluorosis measurement: the use of TF index is appropriate, but not definition os provided for the cutoff (“more than 3”) – clarify the justification.

Authors' response: We thank the reviewer for the suggestion.

Sample selection: As it is a small town, there was no stratification of the sample.

Instruments: We added three studies on the use of EBIA in the Brazilian context or/and with adolescents.

Fluorosis measurement: The issue of dichotomizing dental fluorosis was based on whether the stages from TF=4 to 9 are moderate or severe with the appearance of “pits”.

Pérez-Escamilla R, Segall-Corrêa AM, Kurdian ML, Sampaio MD, Marín-León L, Panigassi G. An Adapted Version of the U.S. Department of Agriculture Food Insecurity Module Is a Valid Tool for Assessing Household Food Insecurity in Campinas, Brazil. J Nutr 2004; 134(8):1923-1928.

Brasil. Secretaria de Avaliação e Gestão da Informação. Escala Brasileira de Insegurança Alimentar – EBIA: análise psicométrica de uma dimensão da Segurança Alimentar e Nutricional. 2014.

Fleitlich-Bilyk B, Goodman R. Prevalence of child and adolescent psychiatric disorders in southeast Brazil. J Am Acad Child Adolesc Psychiatry 2004;43(6):727- 734. https://doi.org/10.1097/01.chi.0000120021.14101.ca

Comment Statistical analysis:

Use of poisson regression with a robust variance is commendable. Still, explain why this was preferred over logistic regression.

Authors' response: We thank the reviewer for this observation. We opted for Poisson regression with robust variance rather than logistic regression due to the high prevalence of the outcome variable—food insecurity—which affected 69.2% of the study population. In such scenarios, where the prevalence of the outcome exceeds 10%, and especially when it surpasses 30%, the odds ratios produced by logistic regression tend to substantially overestimate the strength of association. This can lead to misleading interpretations, particularly when odds ratios are misinterpreted as risk or prevalence ratios. Poisson regression with robust variance, in contrast, allows for the direct estimation of prevalence ratios (PR), which are more appropriate and interpretable in cross-sectional studies with common outcomes. As such, this method provides a more accurate and meaningful measure of association in our study context. We assessed the goodness-of-fit of the Poisson regression model using both the deviance and Pearson chi-square tests.

Comment Mention any multicollinearity checks or goodness-of-fit indicators, is performed. 

Authors' response: The results indicated an excellent fit: deviance goodness-of-fit = 59.00 (p = 1.000) and Pearson goodness-of-fit = 41.63 (p = 1.000). The chi-square test results suggest that there were no statistically significant differences between the observed and expected values, supporting the adequacy of the model's fit to the data.

Comment Discussion

Expand the comparison with other Brazilian or international FI prevalence studies in similar age groups.

Acknowledge the potential role of unmeasured confounders (e.g. family structure stability, dietary patterns)

Discuss the possible bidirectional relationship between poor academic performance and FI more explicitly

Authors' response: Authors' response: We thank the reviewer for the suggestion. As suggested by Reviewer 3, we separated the discussion into blocks and added other national and international studies. We have also broadened the discussion around public policies and intersectorality.

Smith L, López Sánchez GF, Tully MA, Jacob L, Kostev K, Oh H, Butler L, Barnett Y, Shin JI, Koyanagi A. Temporal Trends in Food Insecurity (Hunger) among School-Going Adolescents from 31 Countries from Africa, Asia, and the Americas. Nutrients. 2023 Jul 20;15(14):3226.

Kotchick, B.A.; Whitsett, D.; Sherman, M.F. Food Insecurity and Adolescent Psychosocial Adjustment: Indirect Pathways through Caregiver Adjustment and Caregiver–Adolescent Relationship Quality. J. Youth Adolesc. 2021; 50:89–102.

Baer, T.E.; Scherer, E.A.; Fleegler, E.W.; Hassan, A. Food Insecurity and the Burden of Health-Related Social Problems in an Urban Youth Population. J. Adolesc. Health. 2015; 57:601–607Castro, M. C. et al. Brazil’s unified health system: The first 30 years and prospects for the future. Lancet. 2019;394:10195:345–356.

BRASIL, Ministério da Saúde. Diretrizes da Política Nacional de Saúde Bucal. Brasília, DF: Ministério da Saúde, 2004.

BRASIL. Ministério da Saúde. Lei nº 14.572, de 2023. Brasília, DF: Ministério da Saúde, 2023.

Romão MEDB, Forte FDS, Frazão P, Sampaio FC, Nunes JMFF. Level of natural fluoride in public water supply: geographical and meteorological factors in Brazils Northeast. Braz. oral. Res 2023;37:1-11. https://doi.org/10.1590/1807-3107bor- 2023.vol37.0101

United Nations. Goal 2: Zero Hunger. Available online: https://www.un.org/sustainabledevelopment/hunger/

Kroth DC, Geremia DS, Mussio BR. National School Feeding Program: a healthy public policy. Cien Saude Colet. 2020 Oct;25(10):4065-4076

Baker W. Food Banks in Schools and the ‘Cost of Living’ Crisis; British Educational Research Association: London, UK, 2022.

Comment Limitations

Add potential biases (e.g. social desirability in SDQ answers, recall bias)

Reinforce the limitation of generalizing findings beyond the specific municipality

Autor: We thank the reviewer for the suggestion. We have also added the issues mentioned by Reviewer 2 in the limitations.

Comment Conclusions

Consider restating policy implications more concretely, e.g. the need for school-based nutritional support programs or screening for FI durinf routine health assessments.

Autor: We thank the reviewer for the suggestion. We have restructured the conclusion as suggested.

Comment English language

Minor grammatical issues and awkward phrasing appear sporadically (e.g. Paraíba – inland region of Paraíba)

A light copy-edit by native English speaker is advised to enhance fluency redundancy (e.g. avoid repeating “with FI” multiple times in the same sentence)

Authors' response: The paper has been revised by a specialized company as suggested by the Journal.

Reviewer 3 Report

Comments and Suggestions for Authors

The manuscript analyze the relation between oral health and food insecurity in a sample of 192 young (11-14 years old) in Brazil. My comments:

  1. In the abstract is mentioned "The prevalence of AI was 69.19". What is AI? It needs to be defined before it is used.
  2. The Introduction has many small paragraph that needs to be developed. I would recommend 3-4 paragraphs of 10-12 lines per paragraph (3-4 statements per paragraph). This same guideline can be applied to the rest of the text.
  3. A sentence should not start with a number. Please see the section "Results" and the "Abstract".
  4. Close to 80% make less than the minimum wage, 93% visit a dentist and 43% of dental cavities and 70% of food insecurity. In my impression, the sample has a large proportion of low-income households, and as expected, high prevalence of food insecurity. Given the vulnerable population segments, they have a large proportion that visit the dentist and regular proportion of people with dental cavities. The most unexpected result, in my opinion, is the large proportion of people that visit a dentist. I am not sure that the rest is novel.
  5. Some statements in the "Discussion" are not supported by references, which seems a personal opinion of the authors rather than a discussion based on evidence.

I hope that my comments helps to improve the article. Regards

Author Response

Review 3

  1. Comment In the abstract is mentioned "The prevalence of AI was 69.19". What is AI? It needs to be defined before it is used.

Authors' response: We thank the reviewer for the suggestion. The change was made sentence, change AI for FI.

  1. Comment The Introduction has many small paragraph that needs to be developed. I would recommend 3-4 paragraphs of 10-12 lines per paragraph (3-4 statements per paragraph). This same guideline can be applied to the rest of the text.

Authors' response: We thank the reviewer for the suggestion. The paper has been revised by a specialized company as suggested by the Journal.

  1. Comment A sentence should not start with a number. Please see the section "Results" and the "Abstract".

Authors´response: We thank the reviewer for the suggestion. The paper has been revised by a specialized company as suggested by the Journal.

  1. Comment Close to 80% make less than the minimum wage, 93% visit a dentist and 43% of dental cavities and 70% of food insecurity. In my impression, the sample has a large proportion of low-income households, and as expected, high prevalence of food insecurity. Given the vulnerable population segments, they have a large proportion that visit the dentist and regular proportion of people with dental cavities. The most unexpected result, in my opinion, is the large proportion of people that visit a dentist. I am not sure that the rest is novel.

Authors´response: The municipality has 100% coverage of the Family Health Strategy, which guides primary health care, i.e. there is a family health team made up of doctors, nurses, community health agents and nursing technicians, and an oral health team made up of a dental surgeon and an oral health assistant. The teams carry out various actions in the field of health promotion, education and prevention, as well as providing oral health care at the primary health care level. Brazil has had a National Oral Health Policy since 2004 and the public water supply has natural concentrations of fluoride. In addition, there are federal government programs such as Bolsa Familia and the National School Feeding Program. In the municipality, there is social assistance to support adolescents and their families. We have added this information to the discussion of the article. Thank you for your suggestion.

Castro, M. C. et al. Brazil’s unified health system: The first 30 years and prospects for the future. Lancet. 2019;394:10195:345–356.

BRASIL, Ministério da Saúde. Diretrizes da Política Nacional de Saúde Bucal. Brasília, DF: Ministério da Saúde, 2004.

BRASIL. Ministério da Saúde. Lei nº 14.572, de 2023. Brasília, DF: Ministério da Saúde, 2023.

Romão MEDB, Forte FDS, Frazão P, Sampaio FC, Nunes JMFF. Level of natural fluoride in public water supply: geographical and meteorological factors in Brazils Northeast. Braz. oral. Res 2023;37:1-11. https://doi.org/10.1590/1807-3107bor- 2023.vol37.0101

Kroth DC, Geremia DS, Mussio BR. National School Feeding Program: a healthy public policy. Cien Saude Colet. 2020 Oct;25(10):4065-4076

  1. Comment Some statements in the "Discussion" are not supported by references, which seems a personal opinion of the authors rather than a discussion based on evidence.

Authors´response: We have revised the discussion and included references. We have tried to add the references in such a way as to bring our results into line with those found in other papers. We have also tried to position ourselves in the context analyzed and propose possible ways forward in the face of a complex problem. This requires intersectoral, interdisciplinary and interprofessional efforts between various sectors: education, health, social assistance and justice. We hope we have responded to your suggestion.

Comment I hope that my comments helps to improve the article. Regards

Authors´response: we really appreciat

Round 2

Reviewer 3 Report

Comments and Suggestions for Authors

The manuscript includes most of my previous concerns. 

Author Response

The following justifications have been ordered following the comments of each reviewer:

we added a cross-sectional study to tittle

we removed the "<" at table 2

we added the p-value to table 2 for the dental fluorosis variable

we really appreciate the suggestions and thank you.